# Phytochemicals: Principles and Practice

**DOI:** 10.3390/biology15010018

**Published:** 2025-12-21

**Authors:** Lara Popovic, Ger T. Rijkers

**Affiliations:** Department of Health, Cognition, and Behavior, University College Roosevelt, 4330 AB Middelburg, The Netherlands; larapopovicmail@gmail.com

**Keywords:** phytochemicals, polyphenols, curcumin, artemisinin, bioavailability, metabolism, sustainable medicine

## Abstract

Plants, like all other organisms, evolved chemical defences to protect themselves from predators. These compounds, phytochemicals, come in great variety and have been used in traditional medicine for centuries, with some (such as artemisinin) making their way into modern medicine. Phytochemicals have proven useful in many human applications, from supporting cardiovascular health and managing obesity to aiding wound healing and fighting cancer, yet they remain a largely untapped resource. To maximize their benefits, research must be standardized, and their use carefully regulated and monitored. Progress is slowed, however, by inconsistent legislation, as well as scientific challenges in synthesizing, stabilizing, and standardizing plant compounds. These challenges are compounded by the fact that each plant produces the substances at different rates and that the people that consume phytochemicals metabolize them differently depending on their gut microbiome. This review starts from the basics of phytochemistry and expands to cover plant defense strategies, global healthcare challenges, and the need for sustainable policy, with the goal of helping the field flourish as fully as the plants it studies.

## 1. Introduction

In the 17th-century Netherlands, the Universities of both Amsterdam and Utrecht, within a few years after their foundation, opened a so-called Hortus Medicus: Amsterdam in 1638 and Utrecht in 1639. The initiator of the Hortus Medicus in Amsterdam was Nicolaes Tulp, also known as the central figure in the painting The Anatomical Lesson by Rembrandt van Rhijn [1]. The hortus was a teaching garden for medical students. In Amsterdam, students who had completed full clinical training received a medal as proof of their clinical skills and could obtain a separate medal for their knowledge of medicinal plants [2]. It should be stated that in Italy, those medical botanical gardens were already in place over a century earlier; the Padua Hortus Medicus opened in 1545 [3]. 

The above examples illustrate that knowledge about the medicinal properties of specific plants was well integrated in European medical schools and universities for centuries. Also, extraction procedures and quality control were strictly regulated, although not according to today’s Good Laboratory Practice and Good Manufacturing Practice standards.

### 1.1. Global Healthcare Challenges and Potential Phytochemical Solutions

The past century has witnessed extraordinary progress in medical diagnostics and therapeutics, with biotechnology innovations accelerating particularly between 1991 and 2006 as evidenced by the tripling of related patents [4]. However, these advancements have failed to translate equitably across global populations, creating persistent healthcare disparities. Current data reveals that 90% of worldwide health research funding targets just 10% of the global disease burden, leaving developing nations bearing the greatest burden of preventable diseases [5,6,7]. This imbalance manifests itself in tragic outcomes, including approximately 6 million childhood deaths annually from preventable causes and life expectancies that lag decades behind high-income countries [5]. The limitations of conventional biomedical approaches become strikingly apparent in comparative health system analyses: Germany achieves superior outcomes including higher life expectancy and lower infant mortality while spending 40% less on healthcare per capita than the United States [8]. 

The above sketched landscape has spurred growing interest in sustainable, preventative approaches [9] with phytochemicals emerging as promising solutions to bridge these gaps. Traditional plant-based therapies, utilized by 70–80% of the global population according to anthropological studies [10], offer cost-effective and sustainable interventions with proven efficacy. In a number of cases, phytochemicals have been the basis and starting point for a number of drugs now in use for treatment of cancer, for prevention of rejection of transplanted organs, for treatment of infectious diseases, and other applications. Vinblastine is an anticancer drug derived from indole alkaloids from *Catharanthus roseus* [11]. During the first phase of the COVID-19 pandemic, China’s Lung Cleansing and Detoxifying Decoction, incorporating 21 characterized herbal compounds, was shown to be effective in treatment of infected patients [12]. Likewise, Lung-toxin Dispelling Formula No. 1, known also as Respiratory Detox Shot (RDS), proved effective in mitigating COVID-19 patients’ disease markers. Both failed to meet Western regulations and have not been used in clinical trials, potentially due to their status as dietary supplements in the US and EU, as well as their mechanisms of action not being fully understood [13]. Similarly, artemisinin derivatives from *Artemisia annua*, with centuries of use in traditional Chinese medicine, now form the frontline malaria treatment, saving many lives in Africa where the disease burden is heaviest [14]. Many phytochemical-based drugs, from anticancer drugs like vinblastine to cyclosporin, an immunosuppressive drug discovered serendipitously by mycologists in the 1980s [15,16], used in Western medicine have already been in use for decades. Possibly due to the unclear regulatory status of phytochemicals in pharmacology, research into novel phytochemicals is stagnating. These plant-derived compounds present unique advantages: they require only basic resources like fertile soil, CO_2_ and sunlight for production, making them particularly viable for low-resource settings [17]. Moreover, their multi-target mechanisms of action align well with complex, multifactorial diseases that challenge conventional single-target pharmaceuticals [18].

In a world with rising antimicrobial resistance and chronic disease burdens, exacerbated during the COVID-19 pandemic as resource-limited regions struggled to implement high-tech and high-cost solutions, phytochemicals could offer a valuable mode for both treatment and prevention strategies. Their extensive historical use data, lower production costs, and potentially synergistic bioactive combinations position them as critical tools for equitable healthcare. This becomes particularly evident when integrated with evidence-based policy frameworks like the Ottawa Model of Research Use (OMRU) [19]. The OMRU includes the practice environment, potential adaptors, and health-related outcomes into account as central components in the research use process [19]. The growing body of clinical evidence, from the Nobel Prize-winning validation of artemisinin [20] to recent studies on multi-herb formulations, underscores their potential to address pressing global health inequities when properly standardized and implemented [21].

### 1.2. Botanical Functions and Distribution of Phytochemicals

Phytochemicals serve as essential defence compounds for plants as well as for other interactions with their environment. Simultaneously, phytochemicals can provide significant health benefits when consumed by humans. These bioactive molecules are strategically concentrated in plant organs most vulnerable to predation or environmental stress, particularly in fruits, vegetables, nuts, seeds, and herbs [22]. Their ecological roles are multifaceted—they deter herbivores through bitter tastes or toxic effects, protect against microbial pathogens, and mitigate abiotic stresses like UV radiation [23]. Modern analytical techniques including metabolomics and transcriptomics have allowed us to decipher how plants regulate the production of phytochemicals in response to environmental challenges. The quantitation of stress-induced compounds like flavonoids and alkaloids by LC-MS and GC-MS analyses has been instrumental [24]. The Venus flytrap provides a well-known example of specialized phytochemical deployment in the ecological context, producing compounds that both attract prey and facilitate digestion through secreted enzymes [25]. In edible plants, these defence mechanisms often translate to human health benefits, as seen with glucosinolates from cruciferous vegetables that induce detoxification enzymes in humans [22].

#### 1.2.1. Plant Defence Mechanisms and Pathways

Plants have evolved sophisticated biochemical defence systems utilizing phytochemicals through both direct and indirect mechanisms (Figure 1). Direct antimicrobial action occurs through several strategies including membrane disruption (by compounds like resveratrol), enzyme inhibition (such as catechins blocking chitin synthase), and metal ion chelation [23]. Salicylic acid, jasmonic acid and ethylene mediate signalling pathways involved in the plant responses, both local and systemic responses, against biotic invaders (indirect mechanism). Other plant molecules, including abscisic acid, auxin, and brassinosteroid, modulate these defence responses, either synergistically or antagonistically [26]. Research using mutant plant lines has demonstrated the critical role of specific biosynthetic genes, with PAL- and CHS-deficient mutants showing increased stress susceptibility, while overexpression of genes like stilbene synthase enhances resistance [27]. Remarkably, many plant defence compounds interact with human biological targets in beneficial ways—withanolides from *Withania somnifera* exhibit both anti-herbivore effects and human anti-inflammatory activity [28], while artemisinin’s antimalarial properties derive from its original role as a plant defence compound [29]. This evolutionary conservation of molecular targets underscores the role of these metabolites in protection of plants against internal and external stressors [30].

#### 1.2.2. Structural Classification and Diversity of Phytochemicals

The structural diversity of phytochemicals, estimated to exceed 100,000 distinct compounds [32], reflects their evolutionary adaptation to a myriad of ecological challenges. The chemical classification of phytochemicals organizes these compounds into major categories based on their core structures and biosynthetic pathways, although there is no consensus on a uniform classification system for phytochemicals (Figure 2) [23,33,34]. Because of the structure-based classification, great diversity in function may exist. Polyphenols including flavonoids (quercetin, catechins) and stilbenes (resveratrol) demonstrate potent antioxidant and anti-inflammatory activities, with green tea catechins like EGCG showing particular cardioprotective effects through specific metabolites as illustrated in Table 1 [22]. Terpenoids encompass compounds ranging from volatile monoterpenes to complex carotenoids like lycopene, exhibiting diverse biological activities including antimicrobial effects and vision support [35,36]. Nitrogen-containing alkaloids such as caffeine and morphine demonstrate potent neurological effects in both plants and humans [37,38], while sulphur-containing compounds including glucosinolates release beneficial hydrogen sulphide with antioxidant and anti-inflammatory properties [39]. The metabolic transformation of these compounds, both by bacterial and host enzymes, often enhances their bioactivity, as seen with curcumin which is converted to tetrahydrocurcumin [40]. The bioavailability of curcumin is greatly enhanced by piperine [41]. Within the framework of this paper, it is not possible to discuss all classes of phytochemicals in depth. The reader is referred to excellent reviews on specific classes and subclasses of phytochemicals, including alkaloids [42], carotenoids [43], isoprenoids [44], glucosinolates [45], polyphenols [46], phytosterols [47], polyacetylenes [48], saponins [49], and others such as dietary fibres [50]. Furthermore, subclasses of each main structural class of phytochemicals have been studied individually, such as the subclasses of polyphenols including stilbenes [51], lignans [52], flavonoids [53], ellagic acid [54], and phenolic acid [55]; the flavonoids curcumin [56], anthocyanids [57], flavonols [58], flavones [59], and isoflavones [60]; and the flavanols cathechins [61] and pro(antho)cyanidins [62].

While typical diets provide substantial phytochemical intake, optimal dosing remains challenging to establish due to complex bioavailability considerations that vary significantly between compounds and individuals. It has been calculated that an average French diet gives an intake of 1.6 g of polyphenols per day in summer. In the winter it is 0.2 g lower [63]. In a Finnish study, vegans showed 7–18-fold higher serum concentrations of the polyphenols genistein and daidzein as compared to non-vegetarians [64]. These data show that the intake of phytochemicals can be significantly increased by dietary choices. A daily recommended phytochemical intake has not been established and implemented in guidelines, as has been achieved for other food components such as fibre and sugar. Thus, more studies as the ones cited above would be needed to further substantiate dietary recommendations for phytochemicals. 

**Table 1 biology-15-00018-t001:** Clinically validated phytochemicals: plant sources, health benefits and active metabolites.

Plant Species	Health Benefit	Active Phytochemical	GI Metabolite	References
*Artemisia annua* (Artemisia)	Antimalarial	Artemisinin	Dihydroartemisinin	[20]
*Curcuma longa* (Turmeric)	Anti-inflammatory	Curcumin	Tetrahydrocurcumin	[40]
*Camellia sinensis* (Green tea)	Cardioprotective	EGCG	5-(3′,4′,5′-Trihydroxy-phenyl)-γ-valerolactone	[65]
*Allium sativum* (Garlic)	Antimicrobial	Allicin	Allyl methyl sulfide	[66]
*Vitis vinifera* (Grape)	Antioxidant	Resveratrol	Dihydroresveratrol	[67]
*Zingiber officinale* (Ginger)	Anti-nausea	6-Gingerol	6-gingerol glucuronide	[68]
*Panax ginseng* (Ginseng)	Adaptogenic	Ginsenosides	Compound K	[69]
*Echinacea purpurea* (Echinacea)	Immune stimulation	Alkamides	Tetradeca-8E-10E- dienoic acid isobutylamide	[70]
*Silybum marianum* (Milk thistle)	Hepatoprotective	Silymarin	Sulfated conjugates	[71,72]
*Piper nigrum* (Black pepper)	Bioenhancer	Piperine	Piperonylic acid	[73]

## 2. Principles of Phytochemicals

### 2.1. Foundations and Evolution of Phytochemistry Science 

Phytochemistry represents the interdisciplinary field of study combining botany, organic chemistry, and pharmacology to unravel the complex world of plant metabolites. The field requires close attention to plant systematics and anatomy to ensure accurate sourcing, followed by sophisticated analytical techniques for compound characterization [74]. This is evidenced further by the fact that many phytochemical-based drugs were discovered serendipitously, likely due to the approaches used in such drug discovery programmes [75]. Modern phytochemists employ a suite of chromatographic methods (HPLC, UPLC) coupled with advanced spectroscopy (NMR, MS) to isolate and identify bioactive constituents of candidate species. Solvent selection proves critical, with polar solvents like ethanol optimal for phenolic compounds, while non-polar solvents like supercritical CO_2_ better extract lipophilic terpenes and carotenoids. Computational approaches have revolutionized the field, enabling virtual screening of phytochemical libraries against biological targets and prediction of absorption properties. These fundamental techniques underpin all subsequent applications, from quality control in herbal products to the discovery of novel drug leads. The growing integration of omics technologies (genomics, proteomics) with traditional phytochemical analysis promises to further accelerate our understanding of biosynthetic pathways and their regulation under various environmental conditions [76,77].

### 2.2. Extraction Methods and Technological Advances

The extraction of phytochemicals from plant matrices represents a critical first step in harnessing their therapeutic potential, with methodological choices significantly impacting compound recovery, stability, and subsequent bioavailability [76]. Traditional techniques like maceration and Soxhlet extraction, while still employed, have been largely supplanted by advanced methods offering superior efficiency and selectivity [76]. Solvent-based extractions remain foundational, with polar solvents (methanol, ethanol) optimal for flavonoids and phenolic acids, while non-polar solvents (hexane) better isolate terpenoids and carotenoids—with solvent concentration, duration, and temperature being critical parameters [78]. For volatile compounds such as terpenes and terpenoids, hydrodistillation remains a valuable technique with the added advantage of a low carbon footprint [79]. Ultrasound-assisted extraction (UAE) leverages high-frequency acoustic cavitation to disrupt cell walls, significantly reducing processing time while preserving heat-sensitive compounds like anthocyanins, described for tartaric and malic acids from grapes and winemaking products [80]. This has been demonstrated in UPLC-MS/MS analyses of *Cistus* species, showing optimized recovery of flavonoids, phenolic acids but not anthocyanins [81]. Microwave-assisted extraction (MAE) similarly enhances yields through internal heating and cellular structure rupture, with studies on *Phoenix dactylifera* showing preserved antioxidant activity in date palm constituents extracted at moderate temperatures [82]. 

Supercritical fluid extraction, particularly using CO_2_, has emerged as a green alternative for lipophilic compounds, with adjustable solvation capacity through precise pressure and temperature modulation, enabling selective isolation of terpenes and phytosterols [83]. Enzyme-assisted extraction employs cellulase and pectinase to liberate bound phytochemicals from complex polysaccharide matrices, as evidenced by β-glucosidase treatment enhancing bioactive aglycone yields from soy isoflavones [78]. These advanced methods collectively address key challenges in phytochemical research: maximal phenolic content from *Cistus* can be achieved using methanol-based UAE [81], while EGCG recovery from green tea can be optimized via MAE [76]. Current innovations include 3D intestinal organoid systems for absorption studies and automated high-throughput screening to refine extraction parameters, moving toward standardized preparations for clinical applications as summarized in Table 2 comparing methods for curcumin, resveratrol, and other key compounds. 

**Table 2 biology-15-00018-t002:** Selected phytochemicals: sources, extraction methods, and bioavailability.

Phytochemical	Plant Sources	Extraction Method(s)	Bioavailability Profile
Curcumin	*Curcuma longa*	Solvent Extraction, MAE ^1^, SFE	Low; enhanced by nanoencapsulation and piperine co-administration
Resveratrol	Grapes, Berries	Solvent Extraction, SFE	Moderate; rapid metabolism limits bioactivity
Quercetin	Onions, Apples	Solvent Extraction, UAE	Low; increased when consumed with dietary fats
Catechins	Green Tea	Solvent Extraction, MAE	Moderate; stable in aqueous form, affected by intestinal metabolism
Lycopene	Tomatoes	Solvent Extraction, SFE	Low; better absorbed when cooked with oil
Genistein	Soybeans	Solvent Extraction, Enzyme-Assisted Extraction	Variable; metabolized by gut flora, enhanced by fermented products
Silymarin	Milk Thistle	Solvent Extraction	Poor; improved with phospholipid complexes

^1^ MAE, microwave-assisted extraction; SFE, supercritical fluid extraction; UAE, ultrasound-assisted extraction (based on data from references [84,85,86,87]).

### 2.3. Metabolism, Bioavailability, and Clinical Translation 

The journey from phytochemical consumption to biological activity involves complex metabolic transformations that fundamentally determine therapeutic efficacy, as detailed in Table 3 outlining GI tract processing. Traditional Phase I and II enzymes have impacts on phytochemical metabolism, for example in the bioactivation of simple phenol tyrosol (CAS 501–94–0) into hydroxytyrosol (CAS 10,597–60–1), by activities of isoenzymes of P450 [88]. Phase II metabolism such as sulfation, glucuronide conjugation, or methylation enhances their metabolic activity [89]. In the stomach, acidic conditions release lycopene from food matrices and initiate micelle formation, while the small intestine hosts critical biotransformations, although some not as extensive as those mediated by the liver: curcumin undergoes extensive first-pass metabolism via glucuronidation/sulfation [89], resveratrol is rapidly conjugated [90], and quercetin glycosides are hydrolyzed to absorbable aglycones by lactase-phlorizin hydrolase/cytosolic β-glucosidase (LPH/cBG) enzymes [91,92]. The colonic microbiota then processes non-absorbed compounds through reductive cleavage and dehydroxylation, converting genistein to equol [93], catechins to γ-valerolactones [94], and resveratrol to dihydroresveratrol [95]. This creates a dual metabolic system that challenges conventional pharmacokinetic models [96]. These rules predict the absorption of compounds post oral ingestion and are typically used for toxicology studies in gastroenterology. Determining the biological effect of phytochemicals in vivo at the target site and subsequently on the health of other systems is an important step in promoting the inclusion of phytochemicals. Several studies have outlined specific methodologies for phytochemical absorption. By integration of three classical rules, L-Ro5, Lipinski’s and the Ghose filter, combined with parameters fit to phytochemicals’ physical properties, five phytochemicals within three common sources (ginger, echinacea and tobacco) have been analyzed to establish a protocol for predicting phytochemical metabolite absorption [96]. Although the latter study was well conducted, and others show potential, there has been little change in legislation or methodology in the research of phytochemicals. The different types of secondary metabolites described above are converted through different processes outlined in Table 3 and visualized in Figure 3. 

A substantial fraction of dietary phytochemicals, particularly large and structurally complex compounds such as proanthocyanidins, ellagitannins, and glycosylated flavonoids, escape absorption in the upper GI tract and reach the colon intact. There, they are subjected to microbial catabolism by the resident gut microbiota [97]. This process involves reductive cleavage, dehydroxylation, demethylation and ring fission reactions that convert complex phytochemicals into smaller, absorbable catabolites, also known as gut-derived phenolic metabolites or gut microbial metabolites (GMMs) [98]. 

For example, procyanidins and (epi)catechins undergo stepwise microbial breakdown. The gut bacteria *Eggerthella lenta* rK3 and *Eubacterium* sp. strain SDG-2 are implicated in cleaving the C-ring of (epi)catechins. The intermediate products are further converted by *Flavonifractor plautii* aK2 into 5-(3′,4′-dihydroxyphenyl)-γ-valerolactone (DHPV), a key metabolite also found after consumption of (epi)catechin-rich foods such as green tea and cocoa [99]. DHPV may undergo further catabolism through β-oxidation and α-oxidation into smaller compounds like 3-hydroxyphenylpropionic acid (DHPA) and 3-hydroxyphenylacetic acid (DHAA), which have been identified in plasma and urine samples post-consumption [100].

Additionally, the microbial metabolism of isoflavones such as daidzein leads to the production of equol, a metabolite with higher estrogenic and antioxidant activity than its precursor, but only in individuals hosting specific microbial species such as *Slackia isoflavoniconvertens*. Similarly, lignans from flaxseed are converted to enterolactone and enterodiol by *Clostridium* and *Bacteroides* spp., metabolites linked with hormonal modulation and reduced cancer risk [101].

**Table 3 biology-15-00018-t003:** Metabolism of phytochemicals in the gastrointestinal tract.

Region	Phytochemicals and Metabolism	References
Stomach	Lycopene is released from the food matrix due to low pH and emulsified for micelle formation, facilitating absorption in the small intestine.	[102]
Small Intestine	Curcumin undergoes extensive first-pass metabolism via glucuronidation and sulfation, limiting systemic availability.	[56,103]
	Resveratrol is absorbed here and conjugated into glucuronides and sulfates. Bioavailability of resveratrol is limited by rapid metabolism.	[90,104]
	Quercetin glycosides are hydrolyzed by enzymes like LPH ^1^ and cBG into aglycones that passively diffuse through enterocytes.	[91]
	Catechins such as EGCG are partially absorbed; unabsorbed catechins reach the colon.	[90,91,92,93,94,95,96,102,104,105,106,107]
	Genistein is deglycosylated into aglycones by intestinal enzymes before absorption.	[92,108]
	Silymarin components like silybin are absorbed inefficiently due to poor water solubility; partial metabolism occurs in enterocytes.	[109,110]
Colon (Large Intestine)	Curcumin that escapes upper GI absorption is further reduced and degraded by gut microbes.	[56,111]
	Resveratrol is converted by microbiota into dihydroresveratrol and other phenolics.	[112]
	Quercetin is broken down into smaller phenolic acids such as 3,4-dihydroxyphenylacetic acid by colonic bacteria.	[113,114]
	Catechins are degraded to γ-valerolactones like DHPV by gut microbes as well as intestinal enzymes (e.g., *Flavonifractor plautii*).	[98]
	Genistein is metabolized into equol, a more estrogenic metabolite, by bacteria like *Slackia isoflavoniconvertens* (only in equol producers).	[92]
	Silymarin constituents undergo microbial breakdown into less active phenolics and acids.	[109,110]

^1^ cBG: cellular β-glucosidase, DHPV: 5-(3′,4′-dihydroxyphenyl)-g-valerolactone, EGCG: epigallocatechin-3-gallate, LPH: lactase phlorizin hydrolase.

Collectively, this coordinated human–microbial metabolic interface forms the foundation for understanding the health effects of phytochemical-rich diets. Rather than the native dietary compounds, it is often the resulting conjugated or microbially derived metabolites that act as the true bioactive agents within the body. Thus, identifying and quantifying these metabolites in plasma, urine, and fecal samples has become a critical strategy in nutritional biochemistry and functional food research [115]. 

The clinical translation of these insights requires overcoming significant challenges. Current regulatory frameworks vary globally, from U.S. dietary supplement classifications to China’s drug-like evaluations. Pharmacometabolomics now enables personalized approaches by predicting individual metabolic responses based on microbiome profiles [116], while advanced analytics (HPLC-MS/MS) ensure batch consistency. The integration of Lipinski’s rules with phytochemical-specific parameters provides a framework for predicting absorption, yet legislative recognition of these advances lags [96]. Ultimately, the field is moving beyond studying native phytochemicals to focus on their microbially transformed metabolites as the true mediators of health effects, necessitating comprehensive metabolite profiling in nutritional and clinical research to fully realize phytochemicals’ therapeutic potential. 

## 3. Practice of Phytochemicals 

### 3.1. Biomedical Applications: From Molecular Mechanisms to Global Health Impact

Over 25% of pharmaceuticals used during 1980–2000 were directly derived from plants, while another 25% were chemically altered natural products [117]. Still, large pharmaceutical companies focus more on high-throughput screening of synthetic compounds than on natural products [118]. 

Phytochemicals exhibit remarkable therapeutic potential across, amongst others, diverse medical domains, supported by robust clinical evidence and emerging epigenetic insights (Table 4). In oncology, curcumin’s pleiotropic effects encompass NF-κB inhibition, histone deacetylase-mediated epigenetic silencing of oncogenes, and apoptosis induction [119], while sulforaphane from cruciferous vegetables demonstrates dual functionality in Nrf2 pathway activation and hepatic stem cell proliferation [120,121]. Neurodegenerative research reveals curcumin’s blood–brain barrier permeability for amyloid plaque reduction [122]. Resveratrol can reverse eNOS uncoupling, reducing oxidative stress and improving nitric oxide (NO) availability [123]. Apart from the cardioprotective effect, resveratrol also has neuroprotective effects as shown in animal models [124].

The WHO framework [125] contextualizes phytochemical research within global health systems, documenting plant-based therapies as primary care for 40% of low-resource populations and classifying integration strategies into three evidence tiers: clinically validated botanicals (Tier 1, e.g., metformin’s origin is in galegine from *Galega officinalis* [126]), culturally embedded practices (Tier 2, e.g., *Vernonia amygdalina* use in Nigeria [127]), and emerging candidates (Tier 3, e.g., immunomodulatory mushroom polysaccharides [128]). This structure explains the 68% higher adherence to Tier 2 interventions in native cultural contexts [126] and highlights the therapeutic advantages of traditional preparations like fermented turmeric. The framework also accommodates cutting-edge applications, from antimicrobial adjuvancy (berberine–antibiotic synergies) and metabolic regulation (cinnamaldehyde’s insulin sensitization) to circadian-optimized delivery—sulforaphane administered at 7 a.m. achieves 72% greater Nrf2 activation than evening doses [121]. Epigenetic mechanisms are increasingly recognized as central to efficacy, with plant-rich diets slowing Horvath’s epigenetic clock by 1.74 years and reducing IL-6 by 38% in long-term adherents [129], while personalized approaches now integrate genetic polymorphisms, microbiome profiles, and epigenetic markers to optimize interventions [130].

**Table 4 biology-15-00018-t004:** Role of phytochemicals in plants and their medicinal application.

Phytochemical	Role in Plant/Biosynthesis	Human Medicinal Application
Curcumin	Produced by *Curcuma longa* in response to biotic stress; antimicrobial and wound-healing defence	Anti-inflammatory, anticancer (inhibits NF-κB), antioxidant, used in trials for prostate, colorectal, and breast cancers [21,131]
EGCG ^1^	Flavonoid in green tea; deters herbivores, protects against UV damage	Cardiovascular protection, neuroprotection, weight loss support, activates AMPK, antioxidant and anti-inflammatory [132,133]
Sulforaphane	Formed upon tissue damage in cruciferous vegetables; part of plant defence enzyme system	Induces phase II antioxidant [134] enzymes, epigenetic modulation [135], potential anticancer and neuroprotective roles [136]
Myricetin	Plant-derived flavonol; acts in UV protection and defence against microbes	Synergizes with antibiotics [137], breaks biofilm [138], used against multidrug-resistant *Pseudomonas aeruginosa* [139]
Resveratrol	Stilbene synthesized under pathogen attack or UV stress in grapes	Cardioprotective [140], neuroprotective [141], anti-ageing via SIRT1 activation [142], used in metabolic [143] and cognitive disorder trials [142]
Anthocyanins	Pigments for pollinator attraction and ROS scavenging in high-light environments	Antihypertensive [144], reduce LDL oxidation [108], improve vascular elasticity [145], improve cognitive function [146]
Flavonoids	Broad-spectrum metabolites defending against pathogens, UV, and oxidative stress	Anti-inflammatory [147], reduce CVD risk [148], modulate cytokines and endothelial markers like ICAM/VCAM [149]
Polyphenols	Defensive compounds for herbivore deterrence and pathogen resistance	Improve endothelial function [150], reduce oxidative stress [151], support microbiota balance [152], prevent metabolic syndrome [151]
Hydroxycitric acid	Derived from *Garcinia cambogia* rind; involved in regulation of seed germination and storage metabolism	Anti-obesity, appetite suppressant via ATP-citrate lyase inhibition, reduces fat accumulation [153]
Gingerol	Phenolic compound in ginger for antimicrobial defence	Antidiabetic, anti-inflammatory, improves insulin sensitivity and digestive function [154]
Berberine	Isoquinoline alkaloid involved in allelopathy and microbial inhibition in *Berberis* spp.	Improves glucose metabolism, lowers LDL-C, modulates gut microbiota, used in diabetes and metabolic syndrome [155]
Asiatic acid	Triterpenoid in *Bacopa monnieri*, involved in defence signalling and repair processes	Neuroprotective potential [156,157]
Loliolide	Monoterpenoid lactone from many plants; signals stress and regulates interplant interactions	Antioxidant and potential anti-ageing [158], neuroprotective properties [159]

^1^ EGCG: Epigallocatechin-3-gallate; AMPK: 5’ AMP-activated protein kinase; SIRT1: NAD-dependent deacetylase sirtuin-1; ROS: reactive oxygen species; LDL: low-density lipoprotein; CVD: cardiovascular disease; ICAM/VCAM: inter/vascular cellular adhesion molecule.

### 3.2. Dietary Synergies and Precision Delivery: Beyond Quantity 

The bioactivity of dietary phytochemicals represents a complex interplay between source quality, preparation methods, and individual metabolic factors. Evidence consistently demonstrates that whole food sources outperform isolated supplements. The importance of the food matrix for the bioactivity of phytochemicals is widely recognized [160,161]. This enhanced bioavailability stems from natural and diverse food matrices, composed of all food constituents, but especially whole foods, including fibre, macronutrients, and other micronutrients. These constituents offer effects that facilitate absorption and promote synergistic interactions between different phytochemical classes. Moreover, food conservation and preparation techniques can influence the bioavailability of phytochemicals [27,162]. Phytochemicals in the form of supplements lack this matrix as well as the diversity and therefore synergistic effects. Traditional food preparation methods have evolved to optimize these benefits, with fermentation converting soy isoflavones into more bioactive equol forms, while compounds like piperine in black pepper can increase curcumin absorption by >150% through epigenetic regulation of intestinal transporters [131]. Modern food processing presents both challenges and opportunities, as while conventional thermal methods may degrade sensitive compounds, advanced techniques like freeze-drying can preserve up to 89% of anthocyanins compared to traditional drying approaches [163]. These technological considerations are particularly important given the substantial natural variability in phytochemical content, where phenolic compound levels in different lettuce cultivars can vary as much as 15-fold depending on growing conditions and genetic factors [102]. The Mediterranean diet serves as a paradigm for optimal phytochemical utilization, combining olive oil that enhances absorption of fat-soluble compounds with diverse wild greens providing broad polyphenol profiles and traditional fermented foods that increase isoflavone bioavailability eight-fold [129]. This dietary pattern has been associated with a 30% reduction in cardiovascular mortality [164], mediated through multiple mechanisms including the AMPK-activating effects of garlic-derived allicin and the cumulative action of diverse polyphenols [43,161]. Emerging research in chrononutrition reveals that timing of consumption significantly influences efficacy, with compounds like sulforaphane showing 72% greater Nrf2 activation when consumed in the morning compared to evening intake [165].

The ongoing debate regarding supplements versus whole food sources (of phytochemicals) requires careful consideration, specifically whether only concentrated forms could deliver adequate pharmacological doses of specific compounds [13]. On the other hand, whole foods can provide synergistic combinations that may offer superior long-term benefits through epigenetic modulation, microbiome stabilization, and upregulation of endogenous antioxidant systems [102]. This is particularly relevant given that 30–50% of individuals lack specific gut microbiota necessary for complete phytochemical metabolism [146]. This highlights the need for personalized nutritional approaches that consider both food quality and individual physiological factors [166,167,168]. 

### 3.3. Policy Integration and Sustainable Implementation 

Bridging phytochemical research and healthcare delivery requires addressing multifaceted barriers through evidence-based policy. Regulatory fragmentation persists globally, with still a minority of WHO member states maintaining herbal medicine quality standards [169], while economic disincentives hinder investment in non-patentable natural products. The WHO’s 5-phase integration roadmap offers actionable solutions: (1) national documentation (e.g., Brazil’s 2800-species formulary [170]; (2) standardization (Ghana’s GMP for 37 priority herbs); (3) medical education (India’s mandatory AYUSH curriculum in 72% of schools [171]; (4) insurance coverage (Japan’s reimbursement of 148 Kampo formulas [172]; and (5) real-time monitoring (Malaysia’s blockchain-based HerbWatch [173].

Economic models demonstrate compelling returns: for every 1 USD invested in scaling up actions to address noncommunicable diseases (NCD), there is a return of at least 7 USD in increased employment, productivity, and longer life [169,174]. Although cardiovascular diseases remain the leading cause of death worldwide, dietary phytochemicals have been shown to effectively aid in their management and prevention [175].

## 4. Perspectives

The convergence of traditional knowledge and modern science has illuminated phytochemicals’ extraordinary potential as sustainable solutions to global health challenges. In her Nobel Prize for Physiology or Medicine acceptance speech, Dr. Youyou Tu reflects on the challenges and importance of cooperation between Western medicine and the vast resources offered by traditional medicine [176]. Not only does the stigmatization of traditional medicine and its reduction to pseudoscience hinder potential standardization of traditional medication, but it also limits the possibilities of research performed by modern practitioners into ailments not covered by traditional sources. Climate change has caused a surge in illnesses previously believed to affect only tropical regions [177,178,179]. This will lead to more investment in prevention and treatment of those diseases to which research and sources stemming from experienced practitioners from those tropical regions could contribute.

Apart from investing in research into novel phytochemicals, existing ubiquitous natural foods can have medicinal properties based on the phytochemicals they contain. An example is honey, the world-famous product of bees feeding on plant nectar. The production of honey involves enzymatic and non-enzymatic conversion of mostly sugars, but also phytochemicals, from many different floral sources, and its biochemical and pharmacological activities vary depending on origin and processing [180]. Honey contains a variety of biologically active compounds such as flavonoids, vitamins, antioxidants and hydrogen peroxides (H_2_O_2_) [181,182]. Rock paintings in Spain, dated at 5500 BCE, show people climbing up a ladder to reach a beehive [183]. Honey has been used by many cultures for purposes other than as a natural sweetener. Its medicinal properties have been recognized at least since the Mayans’ time when it was used both through topical application for wound healing [184,185,186] and as a dietary supplement [187].

The long history of phytochemicals (Figure 4) has witnessed many changes and fluctuations in dynamics and perceptions in society: from their early, humble beginnings, when their modulation of the organism was merely incidentally observed and thereafter tracked and applied by experienced elders, shamans, and healers, through the opening of first botanical gardens in medical schools, centralizing what can be considered the beginnings of phytochemistry, into modern medicine. Current research practices and techniques aim to unravel the possibility of using the first observed modulations for prevention and treatment of some of the most common diseases. Although each era had some incredible developments, each era is also marked by the same type of struggles: how to share and distribute the in-depth knowledge which encompasses the complete spectrum of the flora on earth, how to streamline and standardize research being performed in such a way that it reaches both researchers and practitioners. Ultimately, standardization and regulation will have to assure that safe and efficient phytochemicals will benefit the people who need them.

## 5. Conclusions

From their evolutionary origins as plant defence compounds to sophisticated biomedical applications, phytochemicals exemplify nature’s ingenuity in addressing biological threats across kingdoms. Significant progress has been made in understanding their mechanisms and applications. However, realizing their full potential requires overcoming persistent challenges in bioavailability, standardization, and evidence translation. Future research should prioritize clinical validation through properly designed clinical trials, development of innovative delivery systems, and exploration of synergistic combinations. Equally important are policy reforms that create pathways for evidence-based phytochemical therapies to reach patients in need, particularly in underserved populations. As the boundaries between food, medicine, and preventive healthcare continue to blur, phytochemicals can play an increasingly central role in sustainable, personalized healthcare. Their unique combination of ecological sustainability, multi-target efficacy, and cultural acceptability positions them as key players in addressing 21st-century health challenges.

## Figures and Tables

**Figure 1 biology-15-00018-f001:**
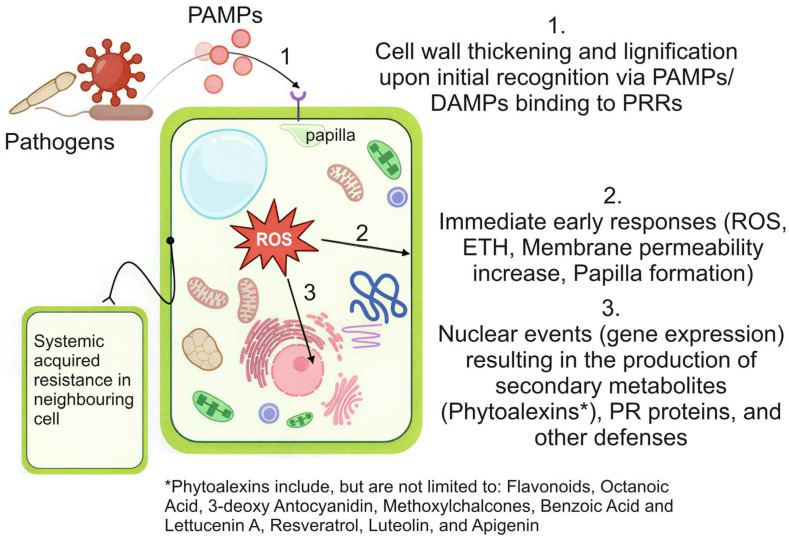
A conceptual model of the plant cell innate immune response triggered by pathogen recognition. The process is divided into three sequential stages: (1) recognition and initial barrier formation, where pathogen-associated molecular patterns (PAMPs) are detected by pattern recognition receptors (PRRs), initiating the first structural defences; (2) signal amplification and early defence, characterized by a burst of reactive oxygen species (ROS), hormonal signalling (e.g., ethylene), and changes in membrane permeability, leading to the reinforcement of physical barriers like the papilla; and (3) induced biochemical defence and systemic immunity, involving the activation of nuclear genes to produce antimicrobial compounds (phytoalexins, pathogenesis-related (PR) proteins) and the priming of neighbouring cells for defence through systemic acquired resistance [31].

**Figure 2 biology-15-00018-f002:**
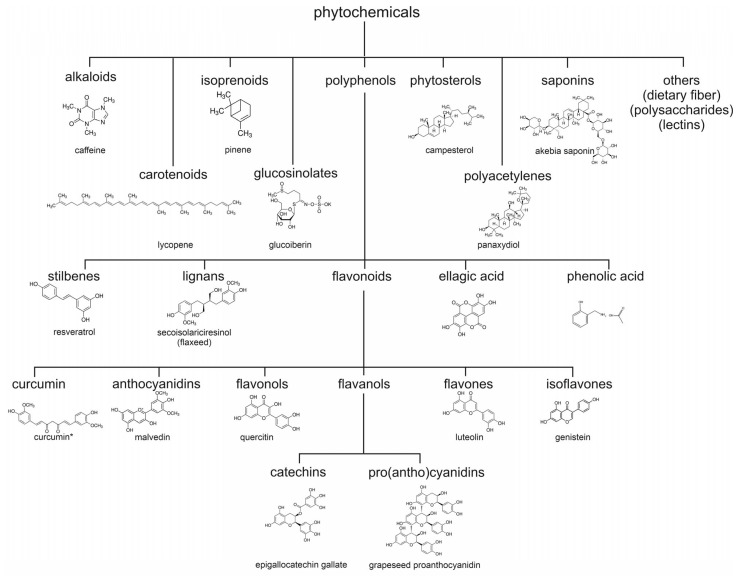
Classification of phytochemicals, based on structural characteristics. Based on references [33,34]. * Please note that curcumin, a polyphenol, is not further classified uniformly.

**Figure 3 biology-15-00018-f003:**
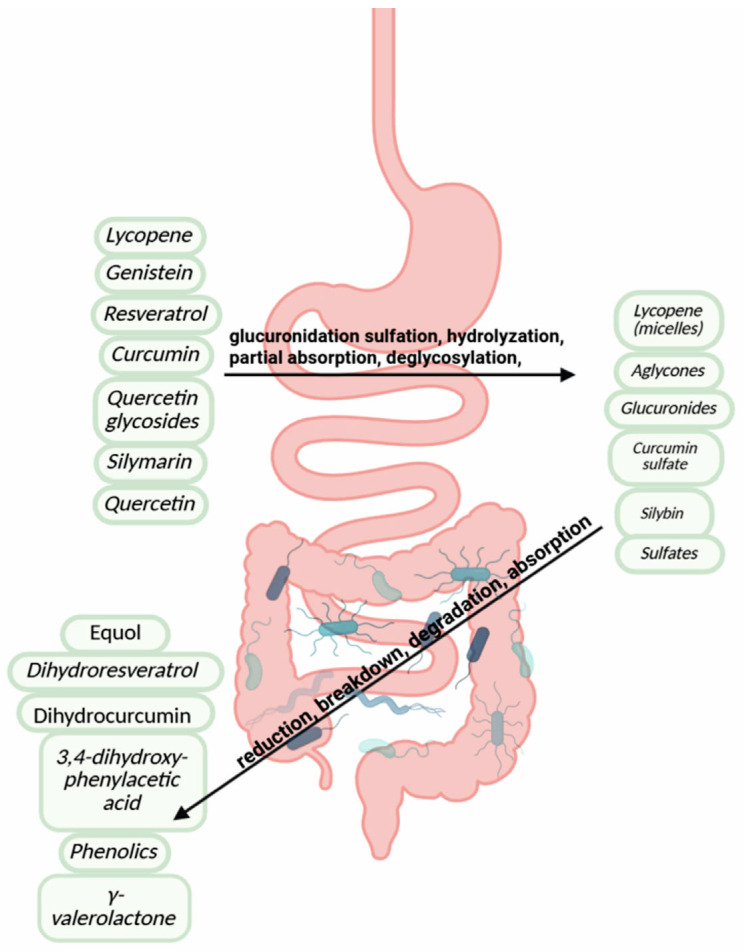
Phytochemical metabolism stages in the gastrointestinal tract.

**Figure 4 biology-15-00018-f004:**
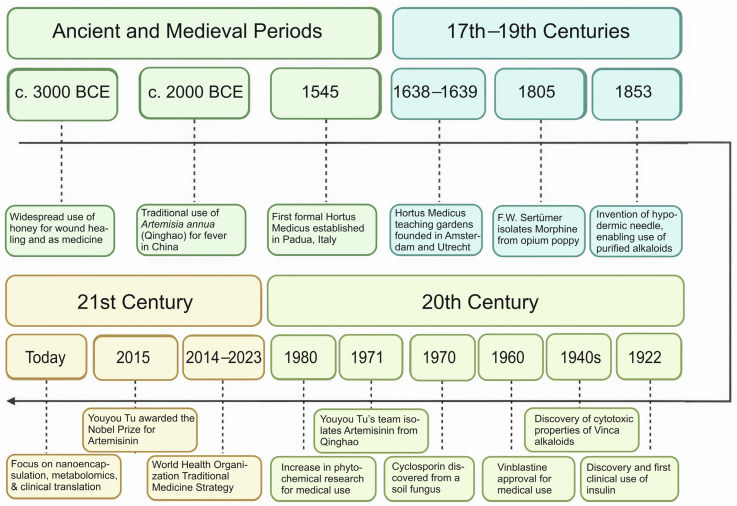
Timeline of milestones of phytochemicals’ discovery and use.

## Data Availability

No new data were created or analyzed in this study. Data sharing is not applicable to this article.

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
