# Peer review of "Phytochemicals: Principles and Practice"

_biology, 2025, doi:10.3390/biology15010018_

Round 1
Reviewer 1 Report
Comments and Suggestions for Authors
The manuscript titled "Phytochemicals: Principles and Practice" provides a ​​comprehensive and in-depth review. Given the growing interest in phytochemicals for biomedical application, it offers valuable insights into the field of medicinal plants. It has the potential to make a valuable contribution to the literature on phytochemical bioactivities.
The manuscript provides a comprehensive analysis of phytochemicals’ critical functions in plant defense and human application for health therapeutics. It examines their potential solutions to address healthcare challenge, structural diversity, immune defense mechanism and ecological significance. Subsequently, it investigates advanced extraction techniques, gastrointestinal and microbial metabolism, bioavailability and innovative delivery systems. Moreover, it analyzes biomedical applications supported by clinical evidence and healthcare delivery based on whole food sources, and discusses policy integration for bridging research and healthcare. Ultimately, the paper presents future research directions, including clinical validation, innovation of delivery systems and investigation of synergistic combinations, alongside the policy reforms for translate evidence-based phytochemical therapies into healthcare.
Here, only a few minor revisions are recommended to enhance the quality and clarity of the manuscript. In my opinion, this manuscript can be accepted for publication after minor corrections.
- Line 78, Change “[12}” as “[12]”.
- Line 79-80, Since “China's Lung Cleansing and Detoxifying Decoction” is considered as a single entity, please change “their” as “its” and “dietary supplements” as “dietary supplement”.
- “Figure1” is not cited in manuscript text, please add a citation at appropriate location.
- Line 236 and Line 355, add “.” after “[12]” and “[103]”. Line 239, add “,” after “[68]”.
- Line 337-340, please change “a natural and diverse food matrixes” as “natural and diverse food matrices” and “all whole foods constituents” as “all of food constituents”.
- Please check the format of all references consistency. Specifically:
- Journal names must be abbreviated and “.” is followed by each abbreviated word, such as in references 69, 73, and 75, while journal names that do not require abbreviation are presented without “.”, such as in references 32, 39, and 69.
- Publication years must be placed after the journal names, such as in reference 85.
- Others, reference 77, change “8(1):” to “8(1),”.
Author Response
The manuscript titled "Phytochemicals: Principles and Practice" provides a comprehensive and in-depth review. Given the growing interest in phytochemicals for biomedical application, it offers valuable insights into the field of medicinal plants. It has the potential to make a valuable contribution to the literature on phytochemical bioactivities.
The manuscript provides a comprehensive analysis of phytochemicals’ critical functions in plant defense and human application for health therapeutics. It examines their potential solutions to address healthcare challenge, structural diversity, immune defense mechanism and ecological significance. Subsequently, it investigates advanced extraction techniques, gastrointestinal and microbial metabolism, bioavailability and innovative delivery systems. Moreover, it analyzes biomedical applications supported by clinical evidence and healthcare delivery based on whole food sources, and discusses policy integration for bridging research and healthcare. Ultimately, the paper presents future research directions, including clinical validation, innovation of delivery systems and investigation of synergistic combinations, alongside the policy reforms for translate evidence-based phytochemical therapies into healthcare.
We thank the reviewer for their thorough analysis and appreciate both the kind remarks as well as the constructive feedback!
Here, only a few minor revisions are recommended to enhance the quality and clarity of the manuscript. In my opinion, this manuscript can be accepted for publication after minor corrections.
Comment 1. Line 78, Change “[12}” as “[12]”.
Reply 1. We have corrected this mistake in notation format
Comment 2. Line 79-80, Since “China's Lung Cleansing and Detoxifying Decoction” is considered as a single entity, please change “their” as “its” and “dietary supplements” as “dietary supplement”.
Reply 2. We have corrected these grammatical mistakes in this example and checked our manuscript again for other instances of grammatical errors.
Comment 3. “Figure1” is not cited in manuscript text, please add a citation at appropriate location.
Reply 3. This was an oversight! We have included the citation after the first sentence of section 1.3
Comment 4. Line 236 and Line 355, add “.” after “[12]” and “[103]”. Line 239, add “,” after “[68]”.
Reply 4. We have added the appropriate punctuation in given lines and checked for similar mistakes.
Comment 5. Line 337-340, please change “a natural and diverse food matrixes” as “natural and diverse food matrices” and “all whole foods constituents” as “all of food constituents”.
Reply 5. We have corrected the first part of the grammatical error. Our original text states specifically that the constituents are found in whole-foods since there is a distinction between whole foods and non-whole foods. Most of the food available to consumers does not fit the description of whole foods and we therefore believe that a distinction should be made. We therefore have changed the corresponding section of the manuscript as follows: “This enhanced bioavailability stems from natural and diverse food matrices, composed of all foods constituents, but especially whole foods, including fibre, macronutrients, and other micronutrients. These constituents offer effects that facilitate absorption and promote synergistic interactions between different phytochemical classes. Moreover, food conservation and preparation techniques can influence the bioavailability of phytochemicals [27, 141].” (Lines 390-394 revised manuscript)
Comment 6. Please check the format of all references consistency. Specifically:
Journal names must be abbreviated and “.” is followed by each abbreviated word, such as in references 69, 73, and 75, while journal names that do not require abbreviation are presented without “.”, such as in references 32, 39, and 69.
Reply 6. We have corrected all errors found in the citations and ensured that all papers are well referenced.
Comment 7. Publication years must be placed after the journal names, such as in reference 85.
Reply 7. We have put the publication and access years in the references in the correct format.
Comment 8. Others, reference 77, change “8(1):” to “8(1),”.
Reply 8. We have corrected general formatting errors in all references including that in reference 77.
Reviewer 2 Report
Comments and Suggestions for Authors
Dear Authors,
Thank you very much for the opportunity to review your article. I find it very interesting, based on current scientific reports from recent years, rich in numerous scientific references, and well-thought-out in terms of its approach to the subject matter and synthesis of the collected scientific content.
I am sending some comments and suggestions for verification and change.
To correction:
Verse 75: Vinblastine and doxorubicin are anticancer drugs derived from indole alkaloids from Catharanthus roseus [11]. – vinblastine is, but doxorubicin was originally made from the bacterium Streptomyces peucetius. – in cited publication vinblastine only was mentioned.
Verse 206: Ultrasound-assisted extraction (UAE) leverages high-frequency 206 acoustic cavitation to disrupt cell walls, significantly reducing processing time while pre- 207 serving heat-sensitive compounds like anthocyanins [54], as demonstrated in UPLC- 208 MS/MS analyses of Cistus species revealing optimized phenolic recovery. – move the footnote to the end of the statement and possibly specify which phenolic: flavonoids, phenolic acids etc. but not anthocyanins.
Verse 308 - Neurodegenerative - 308 research reveals curcumin's blood-brain barrier permeability for amyloid plaque reduction, 309 complemented by resveratrol's sirtuin-mediated mitochondrial enhancement [64, 96]. – verification of citation, comment: 96- Reversal of eNOS uncoupling by resveratrol may be related to the neuroprotective effect, as this mechanism involves reducing oxidative stress and improving nitric oxide (NO) availability, which is important in neuronal defense, but results of this investigation concerned cardioprotection. 116 – it could be cited better.
Table 4 – P. aeruginosa - there is no full name before, hence the suggestion to use the full name;
a proposal to change the text justification or introduce spaces so that individual pieces of content do not overlap.
Reference 138 - no citation of the footnote was found in the text of the article.
Figure 1 – Apigenin, if all names are capitalized
Table 1, Figure 3 – justification (ex. reference [48])
Tetrahydrocurcumin, hexahydrocurcumin - together written
Cistus species, Eubacterium sp. Clostridium and Bacteroides spp., in vivo - italics
To abbreviation: LPH/cBG, HDAC, NfkB, Nrf2, IL-6, DAMPS
Verse 78: patients [12 }. – should be [12]
Verse 290, 308- to remove unnecessary spaces
References: 6,7 to remove spaces, 69 – without hyperlink, dots at the end where missing.
Best regards,
Reviewer
Author Response
Thank you very much for the opportunity to review your article. I find it very interesting, based on current scientific reports from recent years, rich in numerous scientific references, and well-thought-out in terms of its approach to the subject matter and synthesis of the collected scientific content.
We thank the reviewer for these kind remarks.
I am sending some comments and suggestions for verification and change.
To correction:
Comment 1. Verse 75: Vinblastine and doxorubicin are anticancer drugs derived from indole alkaloids from Catharanthus roseus [11]. – vinblastine is, but doxorubicin was originally made from the bacterium Streptomyces peucetius. – in cited publication vinblastine only was mentioned.
Reply 1. Reviewer is fully correct. Doxorubicin is of bacterial origin, and thus formally not a phytochemical. We have revised this sentence which now reads: “Vinblastine is an anticancer drug derived from indole alkaloids from Catharanthus roseus [11]” (lines 80-81 revised manuscript”
Comment 2. Verse 206: Ultrasound-assisted extraction (UAE) leverages high-frequency 206 acoustic cavitation to disrupt cell walls, significantly reducing processing time while pre- 207 serving heat-sensitive compounds like anthocyanins [54], as demonstrated in UPLC- 208 MS/MS analyses of Cistus species revealing optimized phenolic recovery. – move the footnote to the end of the statement and possibly specify which phenolic: flavonoids, phenolic acids etc. but not anthocyanins.
Reply 2. We thank the reviewer for this important addition. We have changed the relevant section of the manuscript as follows: “This has been demonstrated in UPLC-MS/MS analyses of Cistus species, showing optimized recovery of flavonoids, phenolic acids but not anthocyanins [56].” (lines 236-238 revised manuscript).
Comment 2. Verse 308 - Neurodegenerative - 308 research reveals curcumin's blood-brain barrier permeability for amyloid plaque reduction, 309 complemented by resveratrol's sirtuin-mediated mitochondrial enhancement [64, 96]. – verification of citation, comment: 96- Reversal of eNOS uncoupling by resveratrol may be related to the neuroprotective effect, as this mechanism involves reducing oxidative stress and improving nitric oxide (NO) availability, which is important in neuronal defense, but results of this investigation concerned cardioprotection. 116 – it could be cited better.
Reply 2. We had omitted to give a specific reference for the neuroprotective effect of curcumin, which now has been included: Askarizadeh A, Barreto GE, Henney NC, Majeed M, Sahebkar A. Neuroprotection by curcumin: A review on brain delivery strategies. Int J Pharm. 2020 Jul 30;585:119476. doi: 10.1016/j.ijpharm.2020.119476.
Reviewer is also correct that reference 96 only deals with the cardioprotective effect of resveratrol, not its neuroprotective effect. We therefore have included an additional reference, which is a meta-analysis of the neuroprotective effect of resveratrol in rats. Xue R, Gao S, Zhang Y, Cui X, Mo W, Xu J, Yao M. A meta-analysis of resveratrol protects against cerebral ischemia/reperfusion injury: Evidence from rats studies and insight into molecular mechanisms. Front Pharmacol. 2022 Oct 5;13:988836. doi: 10.3389/fphar.2022.988836.
We have reworked this section as follows: Neurodegenerative research reveals curcumin's blood-brain barrier permeability for amyloid plaque reduction [101]. Resveratrol can reverse eNOS uncoupling, reducing oxidative stress and improving nitric oxide (NO) availability [102]. Apart from the cardioprotective effect, resveratrol also has neuroprotective effects as shown in animal models [103]. (lines 357-361 revised manuscript)
Comment 3. Table 4 – P. aeruginosa - there is no full name before, hence the suggestion to use the full name;
Reply 3. Pseudomonas aeruginosa has been spelled in full.
Comment 1. a proposal to change the text justification or introduce spaces so that individual pieces of content do not overlap.
Reply 3. We agree that the spacing between the 2nd and 3rd column of Table 4 was rather narrow and thus have increased the spacing of those columns.
Comment 4. Reference 138 - no citation of the footnote was found in the text of the article.
Reply 4. This was our mistake. We have removed this paper from the list of references.
Comment 5. Figure 1 – Apigenin, if all names are capitalized
Reply 5. We have revised Figure 1 , and capitalized all names of compounds
Comment 6. Table 1, Figure 3 – justification (ex. reference [48])
Reply 6. We checked the reference [48] in Table 1, but it is the correct reference. The justification and layout of the Tables will be checked when preparing for the page proofs of the paper. The same holds for the justification of the Figures.
Comment 7. Tetrahydrocurcumin, hexahydrocurcumin - together written
Reply 7. We have corrected these errors.
Comment 8. Cistus species, Eubacterium sp. Clostridium and Bacteroides spp., in vivo - italics
Reply 8. We have corrected the text formatting and used Italics where required.
Comment 9. To abbreviation: LPH/cBG, HDAC, NfkB, Nrf2, IL-6, DAMPS
Reply 9. We have added these abbreviations to the list of Abbreviations
Comment 10. Verse 78: patients [12 }. – should be [12]
Reply 10. We have removed any faulty brackets from the text
Comment 11. Verse 290, 308- to remove unnecessary spaces
Reply 11. We carefully checked our document for double spaces and have removed them (hopefully all).
Comment 12. References: 6,7 to remove spaces, 69 – without hyperlink, dots at the end where missing.
Reply 12. We have revised all mentioned references and put them in the correct format for the journal. For reference 69 (#74 in the revised version) we have added the doi.
Reviewer 3 Report
Comments and Suggestions for Authors
OVERALL CONSIDERATION
The article entitled “Phytochemicals: Principles and Practice” aimed to provide information about phytochemicals extraction, metabolism, bioavailability, and their applications in human health.
Although being potentially interesting, the article cannot be accepted in its current form. As reported below, this article presents several critical issues, specifically related to the fact that the topics chosen should have been explored in more depth, and the bibliographical references used are inappropriate, often not containing the information reported. The article has a general tone, never providing in-depth and relevant information for the reader. Moreover, several phytochemical compound classes, particularly relevant for human health, are not properly discussed.
Given the hugeness of the "phytochemicals" topic, authors should have included sections dedicated to the main classes (e.g., polyphenols, alkaloids, glucosinolates, etc.), reporting as examples the most representative compounds of each class. This would have provided the reader with a clear and comprehensive overview of the topic. Since this is a review on phytochemicals, it would have been appropriate to classify and discuss them appropriately.
Moreover, also their activity in human health, in the treatment and/or prevention of diseases, is poorly discussed and not properly reported.
All these considerations are also based on the vastity of the scientific literature on this topic. The authors should therefore have conducted bibliographic research more in depth to adequately address it.
Moreover, the aim of this article is not very clear, as well as the gap that the authors are trying to fill.
Below are reported major and minor issues that exemplify the points discussed above.
Major issues
Line 75: Doxorubicin is not discussed in Ref. [11]. Moreover, structurally it is an antracicline and it was at first isolated in Streptomyces peucetius var. caesius and not derived from Catharanthus roseus.
Line 78-80: It is not very clear what the authors mean in this part of the text. Furthermore, I checked the reference [13] and I could not find all the information that the authors reported.
Line 76-80: The products described in Ref. [12] and Ref. [13] are different. Authors should have discussed them properly.
Line 84: Doxorubicin is a compound of bacterial origin, while ciclosporin is of fungal origin. They could not be defined as compounds of plant origin; therefore, I would not discuss them in a review about phytochemicals. Moreover, doxorubicin is not discussed in either [15] or [16].
1.2. Botanical Functions and Distribution of Phytochemicals: This section does not adequately discuss what reported in the section title. The functions of phytochemicals and their distribution are not adequately reported.
Line 112-115: Ref. [24] does not discuss phytochemicals.
1.3. Plant Defence Mechanisms and Pathways: The description of plant defence mechanisms and the pathways involved is very poor.
Line 123-126: Ref. [23] reports the antimicrobial activity of phytochemicals, but does not mention resveratrol or catechins, nor specific mechanisms of action.
Line 126-128: Authors should have better described the role of salicylic acid, jasmonic acid and ethylene pathways in plant resistance.
Line 128-131: Ref. [24] does not discuss what is reported.
Line 138: What is described in Figure 1 has not been covered in paragraph 1.3.
1.4. Structural Classification and Diversity of Phytochemicals: Having assigned this title, the authors should have discussed the structural features underlying the chemical classification of phytochemicals, but this information is absent. The title does not reflect the information reported.
Line 149-150: Ref. [31] does not report this information.
Line 162-164: Ref. [37] does not report this information.
Table 1:
1)Ref. [40] does not report that Dihydroartemisinin is a GI metabolite.
2)Ref. [41] does not mention 5-(3',4',5'-Trihydroxy-phenyl)-γ-valerolactone.
3)Ref. [43] suggests but does not provide definitive information that Dihydroresveratrol is produced in the gastrointestinal tract. Authors can report this information, but they must specify that is not certain, at least in the paper they cited.
4)Ref. [44] does not mention Hexahydro-curcumin. Moreover, this is not a metabolite of 6-gingerol.
5)Ref. [46] does not mention Tetradeca-8E-10E-dienoic acid isobutylamide and does not conclude that the metabolites are formed in the gastrointestinal tract.
6)Ref. [47] does not discuss the metabolism of Silymarin.
7)Ref. [48] does not discuss Piperonylic acid.
2.1. Fundamentals of Phytochemistry: The aim of this paragraph is unclear.
2.2. Extraction Methods and Technological Advances: The different extraction techniques should have been discussed systematically and more in depth, with a greater focus on the choice of extraction parameters, highlighting the advantages and disadvantages of each technique. Furthermore, since this is a review of phytochemicals, the authors should have provided more complete and comprehensive information about the most suitable technique for the class of compounds being extracted.
Line 200-202: Ref. [52] does not discuss either "traditional" extraction techniques such as maceration and Soxhlet, or innovative ones.
Line 202-206: Ref. [53] does not discuss what is reported.
Line 205: Authors should have discussed also hydrodistillation, which is still applied for volatile compounds, such as terpenes and terpenoids.
Line 206-208: Ref. [54] does not discuss UAE.
Line 216-218: Ref. [53] does not describe enzyme assisted extractions.
Line 219-220: Ref. [54] does not include methanol-UAE extraction.
2.3. Metabolism, Bioavailability, and Clinical Translation: While the passage through the gastrointestinal tract is certainly very important, the authors should also have discussed the impact of phase I and II enzymes on phytochemical metabolism. Some phytochemical classes are also overlooked.
Line 234-235: As reported by Ref. [63] itself, glucuronidation and sulfation mainly occur in liver.
Line 235-236: Ref. [65] does not describe the activity of LPH and cBG enzymes.
Line 255-258: The review [86] is focused on quercetin and derivatives and does not mention proanthocyanidins and ellagitannins. However, the article itself reports that not only microbiota, but also human cells produce enzymes necessary to quercetin absorption. Authors should not overlook this aspect. Moreover, given the aim of the paper, this section should also have included the metabolism of other phytochemicals, such as glucosinolates, low molecular weight terpenes, and alkaloids.
Line 271-276: Ref. [85] does not report this information.
3.1. Biomedical Applications: From Molecular Mechanisms to Global Health Impact: The molecular mechanism underlying phytochemical activities is poorly described.
Line 305-306: Ref. [93] does not discuss about curcumin.
Line 322-323: Ref. [101] does not discuss sulforaphane.
3.2. Dietary Synergies and Precision Delivery: Beyond Quantity: It is not very clear what Authors mean with “precision delivery”. Moreover, it would have been appropriate to report in this paragraph the main vegetal sources of phytochemical classes, as well as a systematic discussion of the impact of food processing techniques on their content and bioavailability. Furthermore, considering that the paragraph is entitled "dietary synergies", it would have also been appropriate to discuss in detail more food combinations useful for promoting better activity and/or bioavailability of phytocompounds.
Minor issues
Line 18: Gut microbiome is not the only factor that influences the metabolism of xenobiotics, including phytochemicals.
Line 78: Correct “}” with “]”
Line 83-86: The sentence is unclear.
Line 106: Phytochemicals are secondary metabolites needed by plants not only for defence but also for other interactions with the environment.
Line 123: Authors have not defined what they mean with direct and indirect mechanisms.
Line 165-175: This information is out of context in this paragraph.
Line 209: Genus names must be written in italic.
Line 211: Species nomenclature must be written in italic.
Line 220: Genus names must be written in italic.
Line 258: Why does Ref. [86] appear before [85] in the text?
Line 308: Remove the space before 94.
Line 314: The authors should clarify that metformin is not present as such in Galega officinalis
Line 316: For consistency, I would include Ref. [100] in brackets also in this case.
Author Response
The article entitled “Phytochemicals: Principles and Practice” aimed to provide information about phytochemicals extraction, metabolism, bioavailability, and their applications in human health.
Although being potentially interesting, the article cannot be accepted in its current form. As reported below, this article presents several critical issues, specifically related to the fact that the topics chosen should have been explored in more depth, and the bibliographical references used are inappropriate, often not containing the information reported. The article has a general tone, never providing in-depth and relevant information for the reader. Moreover, several phytochemical compound classes, particularly relevant for human health, are not properly discussed.
Given the hugeness of the "phytochemicals" topic, authors should have included sections dedicated to the main classes (e.g., polyphenols, alkaloids, glucosinolates, etc.), reporting as examples the most representative compounds of each class. This would have provided the reader with a clear and comprehensive overview of the topic. Since this is a review on phytochemicals, it would have been appropriate to classify and discuss them appropriately.
Moreover, also their activity in human health, in the treatment and/or prevention of diseases, is poorly discussed and not properly reported.
All these considerations are also based on the vastity of the scientific literature on this topic. The authors should therefore have conducted bibliographic research more in depth to adequately address it.
Moreover, the aim of this article is not very clear, as well as the gap that the authors are trying to fill.
Below are reported major and minor issues that exemplify the points discussed above.
We thank the reviewer for these critical remarks. Indeed there is a vast literature on phytochemicals. The PubMed database alone already lists over 60,000 papers on phytochemicals with >85% of those from the last 10 years. It therefore is virtually impossible to address all aspects of phytochemical research in depth in a single review.
Below we have addressed all specific comments as well as possible.
Major issues
Comment 1. Line 75: Doxorubicin is not discussed in Ref. [11]. Moreover, structurally it is an antracicline and it was at first isolated in Streptomyces peucetius var. caesius and not derived from Catharanthus roseus.
Reply 1. Reviewer is totally correct that doxorubicin is a bacterial derived compound. We have revised this part of the manuscript and clarified that only vinblastine is the C. roseus derived drug.
Comment 2. Line 78-80: It is not very clear what the authors mean in this part of the text. Furthermore, I checked the reference [13] and I could not find all the information that the authors reported.
Reply 2. We have clarified this part of the text which now reads as follows: “Both failed to meet stricter Western regulations and have not been used in clinical trials, potentially due to their status as dietary supplements in the US and EU, as well as their mechanisms of action being not fully understood [13 ] (lines 85-87of the revised manuscript)
Comment 3. Line 76-80: The products described in Ref. [12] and Ref. [13] are different. Authors should have discussed them properly.
Reply 3. Reviewer is fully correct. We now have expanded this section and addressed the different compounds separately: During the first phase of the COVID-19 pandemic, China's Lung Cleansing and Detoxifying Decoction, incorporating 21 characterized herbal compounds, was shown to be effective in treatment of infected patients [12 ]. Likewise, Lung-toxin Dispelling For-mula No. 1, known also as Respiratory Detox Shot (RDS) proved effective in mitigating COVID-19 patient’s disease markers. Both failed to meet Western regulations and have not been used in clinical trials, potentially due to their status as dietary supplements in the US and EU, as well as their mechanisms of action being not fully understood [13 ]. (lines 81-87 of the revised manuscript).
Comment 4. Line 84: Doxorubicin is a compound of bacterial origin, while ciclosporin is of fungal origin. They could not be defined as compounds of plant origin; therefore, I would not discuss them in a review about phytochemicals. Moreover, doxorubicin is not discussed in either [15] or [16].
Reply 4. We agree, and this was also mentioned by one of the other reviewers, and have removed doxorubicin from the text. We also agree that cyclosporin is formally not a phytochemical (and we explicitly indicate that it was discovered by mycologists). In the “Simple Summary” of our manuscript we (also) have deleted cyclosporin.
Comment 5. 1.2. Botanical Functions and Distribution of Phytochemicals: This section does not adequately discuss what reported in the section title. The functions of phytochemicals and their distribution are not adequately reported.
Reply 5. We agree that the Header doesn’t properly cover the content. In fact, the sections labeled 1.3 and 1.4 in the original manuscript discuss aspects of the function and distribution of phytochemicals. We therefore have adjusted the subsection numbering of 1.3 and 1.4 into 1.2.1 and 1.2.2, respectively.
Comment 6. Line 112-115: Ref. [24] does not discuss phytochemicals.
Reply 6. We have replaced this reference with Commisso, M., et al., Untargeted metabolomics: an emerging approach to determine the composition of herbal products. Comput Struct Biotechnol J, 2013. 4: p. e201301007. that more precisely discusses the compounds which are mentioned in the text.
Comment 7. 1.3. Plant Defence Mechanisms and Pathways: The description of plant defence mechanisms and the pathways involved is very poor.
Reply 7. The detailed comments about the content of this section have been addressed below. This has improved the structure and content of the particular section.
Comment 8. Line 123-126: Ref. [23] reports the antimicrobial activity of phytochemicals, but does not mention resveratrol or catechins, nor specific mechanisms of action.
Reply 8. While ref. 23 does not mention specifically resveratrol it does mention Phenolic compounds, among which is also resveratrol. The text also mentions flavonoid compounds, which includes catechins.
Comment 9. Line 126-128: Authors should have better described the role of salicylic acid, jasmonic acid and ethylene pathways in plant resistance.
Reply 9. We admit that this part was not crystal clear. We have reformulated this section as follows: “Salicylic acid, jasmonic acid and ethylene mediate signalling pathways involved in the plant responses, both local as well as systemic responses, against biotic invaders. Other plant molecules, including abscisic acid, auxin, brassinosteroid, modulate these defence response, either synergistic or antagonistic [26].” (lines 138-141of the revised manuscript).
Comment 10. Line 128-131: Ref. [24] does not discuss what is reported.
Reply 10. We have replaced this reference by Commisso et al. in the revised list of references
Comment 11. Line 138: What is described in Figure 1 has not been covered in paragraph 1.3.
Reply 11. The revised version of the section attempts to better describe the mechanisms shown in Figure 1.
Comment 12. 1.4. Structural Classification and Diversity of Phytochemicals: Having assigned this title, the authors should have discussed the structural features underlying the chemical classification of phytochemicals, but this information is absent. The title does not reflect the information reported.
Reply 12. While it would be tempting to further elaborate on the structural classification of phytochemicals we believe it would be outside of the scope of the review.
Comment 13. Line 149-150: Ref. [31] does not report this information.
Reply 13. We have replaced the reference with a more appropriate one, Petrén et al.
Comment 14. Line 162-164: Ref. [37] does not report this information.
Reply 14. We have rephrased this sentence and replaced reference [37] with a more relevant manuscript. Moreover, we added a specific reference for the role of piperine. The revised section now reads as: “The metabolic transformation of these compounds, both by bacterial as well as host enzymes , often enhances their bioactivity, as seen with curcumin which is converted to tetrahydrocurcumin [37]. The bioavailability of curcumin is greatly enhanced by piperine [38]. (revised manuscript lines 179-182)
Comment 15. Table 1:
Reply 15. We acknowledge that in this Table many reference were not to the point. We have them all replaced with more appropriate references. In the revised manuscript, those references, as well as additional added references, are highlighted in yellow
Comment 16. 1)Ref. [40] does not report that Dihydroartemisinin is a GI metabolite.
Reply 16. We have corrected the reference to the correct source
Comment 17. 2)Ref. [41] does not mention 5-(3',4',5'-Trihydroxy-phenyl)-γ-valerolactone.
Reply 17. Indeed we have replaced reference [41] with the correct reference [Capasso]
Comment 18. 3)Ref. [43] suggests but does not provide definitive information that Dihydroresveratrol is produced in the gastrointestinal tract. Authors can report this information, but they must specify that is not certain, at least in the paper they cited.
Reply 18. We have replaced the reference with a more concrete example
Comment 19. 4)Ref. [44] does not mention Hexahydro-curcumin. Moreover, this is not a metabolite of 6-gingerol.
Reply 19. we have removed the said compound and replaced the reference in order.
Comment 20. 5)Ref. [46] does not mention Tetradeca-8E-10E-dienoic acid isobutylamide and does not conclude that the metabolites are formed in the gastrointestinal tract.
Reply 20. We have replaced the reference with a more suitable source discussing the metabolism rather than native compound
Comment 21. 6)Ref. [47] does not discuss the metabolism of Silymarin.
Reply 21. We have included another reference in the table better describing the metabolism
Comment 22. 7)Ref. [48] does not discuss Piperonylic acid.
Reply 22.Although reference [48] in our original manuscript did include the metabolism of Piper nigrum and pipirine, we have added a more specific reference as suggested
Comment 23. 2.1. Fundamentals of Phytochemistry: The aim of this paragraph is unclear.
Reply 23. We have included this paragraph to give a short overview of the major advances made in phytochemistry as a science. We have changed the header accordingly because, admittedly, the original header did not cover the content. The revised header now is: 2.1 Foundations and evolution of Phytochemistry Science.
Comment 24. 2.2. Extraction Methods and Technological Advances: The different extraction techniques should have been discussed systematically and more in depth, with a greater focus on the choice of extraction parameters, highlighting the advantages and disadvantages of each technique. Furthermore, since this is a review of phytochemicals, the authors should have provided more complete and comprehensive information about the most suitable technique for the class of compounds being extracted.
Reply 24. Please see our responses below to the secific questions and remarks.
Comment 25. Line 200-202: Ref. [52] does not discuss either "traditional" extraction techniques such as maceration and Soxhlet, or innovative ones.
Reply 25. We have corrected our mistake and replaced reference [52] with a more appropriate one.
Comment 26. Line 202-206: Ref. [53] does not discuss what is reported.
Reply 26. We have replaced this reference with a more appropriate publication (Streimikyte 2022; reference 53 in the revised manuscript)
Comment 27. Line 205: Authors should have discussed also hydrodistillation, which is still applied for volatile compounds, such as terpenes and terpenoids.
Reply 27. We have included hydrodistilation in the revised manuscript. “For volatile compounds such as terpenes and terpenoids, hydrodistilation remains a valuable technique with an added advantage of a low carbon footprint [54]. (lines 231-232 revised manuscript)
Uhl A, Knierim L, Höß T, Flemming M, Schmidt A, Strube J. Autonomous Hydrodistillation with a Digital Twin for Efficient and Climate Neutral Manufacturing of Phytochemicals. Processes. 2024; 12(1):217. https://doi.org/10.3390/pr12010217
Comment 28. Line 206-208: Ref. [54] does not discuss UAE.
Reply 28. We have replaced reference [54] with a source that better describes the method
Comment 29. Line 216-218: Ref. [53] does not describe enzyme assisted extractions.
Reply 29. We have replaced reference [53] with a source that better describes the method
Comment 30. Line 219-220: Ref. [54] does not include methanol-UAE extraction.
Reply 30. Also here, we have replaced the reference with a source that better describes the method
Comment 31. 2.3. Metabolism, Bioavailability, and Clinical Translation: While the passage through the gastrointestinal tract is certainly very important, the authors should also have discussed the impact of phase I and II enzymes on phytochemical metabolism. Some phytochemical classes are also overlooked.
Reply 31. In the revised manuscript, we have included additional text (and references) on the impact of phytochemical metabolism: “Traditional Phase I and II enzymes have impact on phytochemical metabolism for ex-ample in the bioactivation of simple phenol tyrosol (CAS 501–94–0) into hydroxytyrosol (CAS 10,597–60–1) by activities of isoenzymes of P450 [63]. Phase II metabolism such as sulfation, glucuronide conjugation, or methylation enhances their metabolic activity [64]. (revised manuscript lines 265-2690
Comment 32. Line 234-235: As reported by Ref. [63] itself, glucuronidation and sulfation mainly occur in liver.
Reply 32. Reviewer is fully correct and we have changed we have changed the relevant sentence to reflect this. The revised section now reads as: . . . . while the small intestine hosts critical biotransformations, although for some not as extensive as those mediated by the liver: curcumin undergoes extensive first-pass metabolism via glucuronidation/sulfation [67], resveratrol is rapidly conjugated [68], and quercetin glycosides are hydrolyzed to absorbable aglycones by lactase-phlorizin hydrolase/cytosolic β-glucosidase (LPH/cBG) enzymes [69, 70] (revised manuscript lines 270-275).
Comment 33. Line 235-236: Ref. [65] does not describe the activity of LPH and cBG enzymes.
Reply 33. We have added references [69, 70] that describes the activities of the two enzymes in detail.
Comment 34. Line 255-258: The review [86] is focused on quercetin and derivatives and does not mention proanthocyanidins and ellagitannins. However, the article itself reports that not only microbiota, but also human cells produce enzymes necessary to quercetin absorption. Authors should not overlook this aspect. Moreover, given the aim of the paper, this section should also have included the metabolism of other phytochemicals, such as glucosinolates, low molecular weight terpenes, and alkaloids.
Reply 34. In the first section of 2.3 both human enzyme activity as well as microbial metabolism is described. Table 3 provides specific examples of these processes for a number of selected phytochemicals, including references.
Comment 35. Line 271-276: Ref. [85] does not report this information.
Reply 35. We have replaced this reference with a more appropriate one.
Comment 36. 3.1. Biomedical Applications: From Molecular Mechanisms to Global Health Impact: The molecular mechanism underlying phytochemical activities is poorly described.
Reply 36. We have revised this section, also in response to comments from another reviewer. “Neurodegenerative research reveals curcumin's blood-brain barrier permeability for amyloid plaque reduction [101]. Resveratrol can reverse eNOS uncoupling, reducing oxidative stress and improving nitric oxide (NO) availability [102]. Apart from the car-dioprotective effect, resveratrol also has neuroprotective effects as shown in animal models [103].” (lines 357-361 revised manuscript)
Comment 37. Line 305-306: Ref. [93] does not discuss about curcumin.
Reply 37. We have corrected the unfortunate mix-up of references
Comment 38. Line 322-323: Ref. [101] does not discuss sulforaphane.
Reply 38. We have replaced this reference with a more appropriate one.
Comment 39. 3.2. Dietary Synergies and Precision Delivery: Beyond Quantity: It is not very clear what Authors mean with “precision delivery”. Moreover, it would have been appropriate to report in this paragraph the main vegetal sources of phytochemical classes, as well as a systematic discussion of the impact of food processing techniques on their content and bioavailability. Furthermore, considering that the paragraph is entitled "dietary synergies", it would have also been appropriate to discuss in detail more food combinations useful for promoting better activity and/or bioavailability of phytocompounds.
Reply 39. In our manuscript we have mentioned the vegetal nutrient sources at several places, although only briefly. The issue of food combinations was not specifically discussed, however. We have revised other parts of the text to better emphasize consumption of whole food. We have also included a reference to a special issue of Foods Journal [Sultanbawa Y, Netzel ME. Introduction to the Special Issue: Foods of Plant Origin. Foods. 2019 Nov 6;8(11):555. Reference #141] which we find to be a valuable description of food processing techniques and their influence on bioavailablity of phytochemicals.
Minor issues
Comment 40. Line 18: Gut microbiome is not the only factor that influences the metabolism of xenobiotics, including phytochemicals.
Reply 40. We have included the additional information
Comment 41. Line 78: Correct “}” with “]”
Reply 41. We have reformatted the brackets
Comment 42. Line 106: Phytochemicals are secondary metabolites needed by plants not only for defence but also for other interactions with the environment.
Reply 42. Indeed, we have been too restrictive in our description of the function of phytochemicals in plants. We have followed your suggestions and rephrased the sentences as follows: “Phytochemicals serve as essential defence compounds for plants as well as for other interactions with their environment. Simultaneously, phytochemicals can provide significant health benefits when consumed by humans” (lines 115-117 of the revised manuscript).
Comment 43. Line 123: Authors have not defined what they mean with direct and indirect mechanisms.
Reply 43. We have revised this section of our manuscript as follows: “Plants have evolved sophisticated biochemical defence systems utilizing phytochemicals through both direct and indirect mechanisms (Figure 1). Direct antimicrobial action occurs through several strategies including membrane disruption (by compounds like resveratrol), enzyme inhibition (such as catechins blocking chitin synthase), and metal ion chelation [23]. Salicylic acid, jasmonic acid and ethylene mediate signalling pathways involved in the plant responses, both local as well as systemic responses, against biotic invaders (indirect mechanism). Other plant molecules, including abscisic acid, auxin, brassinosteroid, modulate these defence response, either synergistic or antagonistic [26].” (revised manuscript lines 134-141)
Comment 44. Line 209: Genus names must be written in italic.
Reply 44. All genus names now are written in Italic
Comment 45. Line 211: Species nomenclature must be written in italic.
Reply 45. All species names now are written in Italic
Comment 46 Line 220: Genus names must be written in italic.
Reply 46. All genus names now are written in Italic
Comment 47. Line 258: Why does Ref. [86] appear before [85] in the text?
Reply 47. This has a technical reason. We had already referred to Table 3 in the main text. In Table 3 the references have been included. So, reference 85 is in Table 3. When the main text continues, the first new reference included is reference 86. Later in the text we refer back to reference 85. Please not that in the revised manuscript these numbers have changed.
Comment 48. Line 308: Remove the space before 94.
Reply 48. Space before 94 has been removed
Comment 49. Line 314: The authors should clarify that metformin is not present as such in Galega officinalis
Reply 49. The reviewer is correct, we were not clear on the relation between metformin and Galega officinalis. We have rephrased this sentence as follows: Metformin, based on the natural product galegine from Galega officinalis. (revised manuscript line 365)
Comment 50. Line 316: For consistency, I would include Ref. [100] in brackets also in this case.
Reply 50. We have followed this advice and changed this line as follows: (Tier 3, e.g., immunomodulatory mushroom polysaccharides [100]). (revised manuscript lines 367-368)
Round 2
Reviewer 3 Report
Comments and Suggestions for Authors
While I appreciate the effort the authors have put into addressing some of the issues, unfortunately my major concerns regarding this article remain. Specifically, I believe the topics should have been covered in more depth (connected with this, often the title often does not reflect the content of what is discussed in the paragraphs), many classes of phytochemicals are not adequately discussed, and the article does not appear to provide relevant information for the reader.
In my opinion, the article should be restructured in a more detailed, systematic, and comprehensive manner to be relevant to readers interested in the topic.
Therefore, my advice is that authors resubmit this article after carefully rewriting it.
Author Response
Comments 1. While I appreciate the effort the authors have put into addressing some of the issues, unfortunately my major concerns regarding this article remain.
Response 1. In the first round of review, the reviewer had 50 comments. We have done our very best to address them all, not just some of the issues. We therefore regret that there are major concerns remaining.
Comments 2. Specifically, I believe the topics should have been covered in more depth (connected with this, often the title often does not reflect the content of what is discussed in the paragraphs).
Response 2. We agree that each of the paragraphs could be extended, to a degree that each paragraph could be the topic of a dedicated review paper. A more in depth discussion of each topic in our current manuscript would (in our view) not be fitting within the limits of a general review paper.
In review round 1, also a comment was made on the Titles and content of certain paragraphs. Our response to that comment is pasted below.
1.2. Botanical Functions and Distribution of Phytochemicals: This section does not adequately discuss what reported in the section title. The functions of phytochemicals and their distribution are not adequately reported.
We agree that the Header doesn’t properly cover the content. In fact, the sections labeled 1.3 and 1.4 in the original manuscript discuss aspects of the function and distribution of phytochemicals. We therefore have adjusted the subsection numbering of 1.3 and 1.4 into 1.2.1 and 1.2.2, respectively.
Comments 3. Many classes of phytochemicals are not adequately discussed
Response 3. With over a 1000 phytochemicals having been defined as of now, a structure based classification scheme still has not been firmly established. We have included an extra Figure in our manuscript with an overview of the classification and subclassification of phytochemicals. For each class and subclass an example with structural formula is given. The classification is based on Kumar et al (reference #23 in the revised manuscript) as well as Pawasse et al (reference #24 in the revised manuscript).
Kumar A, P N, Kumar M, Jose A, Tomer V, Oz E, Proestos C, Zeng M, Elobeid T, K S, Oz F. Major Phytochemicals: Recent Advances in Health Benefits and Extraction Method. Molecules 2023, 28(2), 887. doi: 10.3390/molecules28020887.
Pawase PA, Goswami C, Shams R, Pandey VK, Tripathi A, Rustagi S, Darshan G. A conceptual review on classification, extraction, bioactive potential and role of phytochemicals in human health. Future Foods 2024, 9, 100313. doi.org/10.1016/j.fufo.2024.100313.
Furthermore, we have added a statement directing the readers to recent, dedicated review papers on specific classes of phytochemicals. “Within the framework of this paper it is not possible to discuss all classes of phyto-chemicals in depth. The reader is referred to excellent reviews on specific classes and subclasses of phytochemicals, including alkaloid [41], carotenoids [42], isoprenoids [43], glucosinolates [44], polyphenols [45], phytosterols [46], polyacetylenes [47], saponins [48], and others such as dietary fibres [49]. Furthermore subclasses of each main struc-tural class of phytochemicals have been studied individually, such as the subclasses of polyphenols: stilbenes [50], lignans [51], flavonoids [52], ellagic acid [53], phenolic acid [54], and the flavonoids curcumin [55], anthocyanids [56], flavonols [57], flavones [58], isoflavones [59] and the flavanols cathechins [60] and pro(antho)cyanidins [61].” (Lines 184-193 of the re-revised manuscript). The 20 additional references have been added.
Comments 4. The article does not appear to provide relevant information for the reader.
Response 4. We are sorry that the reviewer is of the opinion that our manuscript doesn’t provide relevant information. In the reviewer’s summary (a Likert scale from 1 to 5 stars), the other 2 reviewers rated the question “Is the work a significant contribution to the field?” with 5/5 and 4/5 stars, respectively.
Comments 5. In my opinion, the article should be restructured in a more detailed, systematic, and comprehensive manner to be relevant to readers interested in the topic.
Response 5. We a sorry that this reviewer is of the opinion that our manuscript should be restructured into a more detailed, systematic and comprehensive review. We trust the reviewer is a major expert in the field of phytochemicals, as we trust the 2 other reviewers of our manuscript are.
The other reviewers have a completely different view of our manuscript. The quote reviewer #1 “ The manuscript titled "Phytochemicals: Principles and Practice" provides a comprehensive and in-depth review. Given the growing interest in phytochemicals for biomedical application, it offers valuable insights into the field of medicinal plants. It has the potential to make a valuable contribution to the literature on phytochemical bioactivities.
The manuscript provides a comprehensive analysis of phytochemicals’ critical functions in plant defense and human application for health therapeutics. It examines their potential solutions to address healthcare challenge, structural diversity, immune defense mechanism and ecological significance. Subsequently, it investigates advanced extraction techniques, gastrointestinal and microbial metabolism, bioavailability and innovative delivery systems. Moreover, it analyzes biomedical applications supported by clinical evidence and healthcare delivery based on whole food sources, and discusses policy integration for bridging research and healthcare. Ultimately, the paper presents future research directions, including clinical validation, innovation of delivery systems and investigation of synergistic combinations, alongside the policy reforms for translate evidence-based phytochemical therapies into healthcare.”
And to quote reviewer #2: “Thank you very much for the opportunity to review your article. I find it very interesting, based on current scientific reports from recent years, rich in numerous scientific references, and well-thought-out in terms of its approach to the subject matter and synthesis of the collected scientific content.”
Comments 6. Therefore, my advice is that authors resubmit this article after carefully rewriting it.
Response 6. We have tried our best to address the major concerns of the reviewer by adding additional content.